# Immunological Profiles in Parry–Romberg Syndrome: A Case–Control Study

**DOI:** 10.3390/jcm13051219

**Published:** 2024-02-21

**Authors:** Irma Saulle, Antonio Gidaro, Mattia Donadoni, Claudia Vanetti, Alessandra Mutti, Maria Eva Romano, Mario Clerici, Chiara Cogliati, Mara Biasin

**Affiliations:** 1Department of Biomedical and Clinical Sciences, University of Milan, 20157 Milan, Italy; irma.saulle@unimi.it (I.S.); claudia.vanetti@unimi.it (C.V.); chiara.cogliati@unimi.it (C.C.); 2Department of Pathophysiology and Transplantation, University of Milan, 20122 Milan, Italy; mario.clerici@unimi.it; 3Department of Rheumatology, Luigi Sacco Hospital, 20157 Milan, Italy; antonio.gidaro@asst-fbf-sacco.it (A.G.); donadoni.mattia@asst-fbf-sacco.it (M.D.); mutti.alessandra@asst-fbf-sacco.it (A.M.); romano.mariaeva@asst-fbf-sacco.it (M.E.R.); 4Don C. Gnocchi Foundation, IRCCS, 20122 Milan, Italy

**Keywords:** Parry–Romberg, immunological profile, Th17

## Abstract

**Background**: Parry–Romberg syndrome (PRS) is a rare craniofacial disorder. The aim of this study is to provide information on the immunological profile of this pathology. Since PRS can be included in a wider spectrum of sclerodermic diseases, we propose a case–control study comparing a patient affected by PRS with one with a diagnosis of scleroderma, herein used as control (CTR). **Methods**: B lymphocyte, T lymphocyte, and monocyte phenotypes and functions were assessed by flow cytometry in influenza (Flu)- or anti cluster differentiation (CD)3/CD28-stimulated peripheral blood mononuclear cells (PBMCs). Cytokine concentration was evaluated as well in PBMC supernatants, plasma, and saliva by Luminex assay. **Results**: T and B lymphocytes were similarly activated in unstimulated PRS and CTR cells but differed following antigen stimulation. T helper (Th)17 lymphocytes were expanded in PRS compared to CTR; this increase correlated with higher interleukin (IL)-17 concentration. **Conclusions**: Our case–control study is the first to compare the immunological profiles of PRS and scleroderma patients. The higher percentage of Th17 cells in PRS suggests the use of anti-IL17 receptor monoclonal antibody in this rare disease; however, further studies with larger numbers of patients are needed to confirm our findings.

## 1. Introduction

Parry–Romberg syndrome (PRS) is a rare craniofacial disorder affecting approximately 1 in 250,000 people, characterized by hemifacial atrophy of the skin, subcutaneous tissue, fat, and, in severe cases, the underlying muscles and bones [1].

Usually, disease onset is within the first two decades of life and it is characterized by a progressive and variable hemifacial loss of soft tissue, which can extend deeper to osseocartilaginous tissues [2]. Atrophy and deformation of the face progress very slowly from 2 to 20 years, leading to hemifacial atrophy, after which a stage of stabilization is reached. This is usually associated with enophthalmos on the affected side, lingual atrophy, and deviation of the nose and mouth toward the diseased side [2,3,4]. Other clinical features include extracutaneous disease manifestations and neurologic, ocular, and oral pathology, which may present at any stage of the disease [3]. Among neurological manifestations, headaches and seizure disorders are the most common. Seizures (simple or complex partial type) typically arise from the cerebral cortex of the same side and are often resistant to treatment. Neuropathies involving several cranial nerves (the third, fifth, sixth, and seventh) have been described as well. Impingement of the trigeminal nerve due to vascular inflammation and destruction of surrounding bone causes secondary trigeminal neuralgia in these patients [4,5].

The etiology of PRS remains unclear. Potential causes include infection, trauma, sympathetic cervical ganglion dysfunction, abnormal embryogenesis, and vascular abnormalities [6]. Notably, PRS is often described as an autoimmune condition potentially similar to localized scleroderma *en coupe de sabre* (ECDS), a rare variant of localized scleroderma involving the frontoparietal face and skull [7]. This is supported by findings of inflammatory histopathology, serum autoantibodies [8], coexistent autoimmune diseases (e.g., lupus), and positive response to immunosuppression in patients with PRS [3]. Age of onset, associated neurologic symptoms, and cutaneous presentations are other characteristics shared between the two pathologies [9]. On histopathology, both diseases show dermal sclerosis with thickened collagen bundles, chronic lymphocytic infiltrate, and atrophy of adnexa [2]. The most important clinical features distinguishing PRS from ECDS include paramedian atrophy in PRS without significant skin induration and associated atrophy with a tendency to extend down to the face with mandibular and orodental involvement [3]. Despite affecting different parts of the body and presenting with distinct symptoms, these two diseases share underlying pathological mechanisms related to autoimmunity and tissue fibrosis. Based on these premises, we present the results of a case–control study in which immunological parameters assessed in the oral mucosa and in peripheral blood mononuclear cells (PBMCs) of a PRS patient were compared to those of a patient affected by scleroderma (CTR). The clinical features of the two disease are described in Table 1.

Comparing these two conditions could help clinicians to understand the spectrum of autoimmune diseases and to develop better strategies for diagnosis, management, and treatment. Additionally, studying the immunological similarities and differences between PRS and CTR may provide insights into the pathogenesis of autoimmune disorders, leading to the identification of specific biomarkers that could possibly be exploitable to promptly diagnose PRS and/or as new therapeutic targets.

## 2. Materials and Methods

### 2.1. Patients’ Description

The patient affected by PRS is a 45-year-old woman with juvenile onset of PRS and left hemispheric facial atrophy that led to five facial plastic surgeries and symptomatic left frontotemporal epilepsy. After a slow regression over time, the disease relapsed at the age of 30 with uncontrolled seizures, requiring a multiple anti-epileptic drug therapy consisting of valproic acid, clobazam, and lamotrigine. There are no additional notable comorbidities, except for previous anti-viral therapy for the eradication of HCV. At the time of enrollment, she had never taken immunosuppressive drugs.

The control (scleroderma) patient is a 47-year-old woman with a history of 13 years of limited scleroderma, positivity for anti-Scl70 antibodies, Raynaud phenomenon, progressive pulmonary involvement (spirometry: TLC 68%, FEV1/FVC 72%, DLCO 58%), and no other significant comorbidities. At the time of enrollment in our study, she had never taken immunosuppressive drugs and only underwent treatment with intravenous infusions of prostaglandin I_2_ analog (iloprost). A month after blood and salivary sampling, due to symptomatic progression of dyspnea, she started therapy with mycophenolate mofetil, which was subsequently switched to nintedanib, due to lack of tolerance.

### 2.2. Saliva Plasma and PBMC Isolation

Participants were asked not to eat, drink, or smoke for at least 30 min prior to saliva collection. Saliva was incubated at 56 °C for 10 min and centrifuged at 6000× *g* for 10 min. Supernatants were stored at −80 °C until use. Plasma was obtained by centrifugation of whole blood at 1200× *g* for 10 min and stored at −20 °C until use.

PBMCs were isolated from whole blood by density gradient centrifugation using Ficoll (Cedarlane Laboratories Limited, Hornby, ON, Canada), as previously described [10], and viable cells were counted with the cell counter ADAM-MC (Digital Bio, NanoEnTek Inc., Seoul, Republic of Korea).

### 2.3. Cell Culture Conditions

PBMCs were resuspended at a concentration of 1 × 10^6^ PBMCs/mL in RPMI 1640 medium (Euroclone, Milan, Italy) containing 10% fetal bovine serum (FBS), 1% levo-glutammin LG, and 2% penstreptomicin. To evaluate the global responsiveness and functionality of the immune system of both patients, 1 × 10^6^ PBMCs were stimulated with recall antigens (flu vaccine) and 1 μg/mL of anti-CD3 (Invitrogen, Waltham, MA, USA)/CD28 (Biosigma, Venice, Italy). In particular, flu stimulus resulted from the mixing of two UV-inactivated influenza viruses: an influenza A virus (A/RX73 and A/Puerto Rico/8/34 strains; 1:800) and the 1998–1999 formula of flu vaccine (1:5000; Wyeth Laboratories Inc., Marietta, PA, USA). Unstimulated PBMCs were cultured as control (Mock). Cells were harvested 10 h post treatment for flow cytometry and secretome analysis.

### 2.4. Flow Cytometry Analysis

The immunophenotype of lymphocyte subpopulations was investigated by flow cytometry on PBMCs upon anti-CD3/CD28 and flu stimulation. The gaiting strategies of the main lymphocytes’ subpopulations are shown in Appendix A.

**B lymphocytes** were identified as CD45+ (KO525, Beckman Coulter, Milan, Italy), CD20+ (APC Alexa Fluor 750, Beckman Coulter, Milan, Italy), and CD19+ (FITC, Beckman Coulter). B cell subpopulations were further analyzed as follows: plasmablasts, CD38+ (PE-C5.5, Beckman Coulter), CD27+ (PE, Beckman Coulter); transitional cells, CD24+ (ECD, Beckman Coulter); naïve B cells, CD27—(PE, Beckman Coulter), IgD+ (APC, Beckman Coulter); switched-memory B lymphocytes, CD27+ (PE, Beckman Coulter), IgD—(APC, Beckman Coulter); unswitched-memory B lymphocytes, CD27+ (PE, Beckman Coulter), IgD + (APC, Beckman Coulter).

The following **CD4+ and CD8+ T cell subsets were analyzed**: T helper (Th) 17 lymphocytes were identified as CD45+ (KO525, Beckman Coulter), CD4+ (PE-Cy7, Beckman Coulter), IL-17A+ (FITC, Biolegend, San Diego, CA, USA), RORγT+ (PE, eBiosciences, San Diego, CA, USA); T regulatory (Treg) lymphocytes were identified as CD45+ (KO525, Beckman Coulter), CD4+ (PE-Cy7, Beckman Coulter), CD25+ (ECD, Beckman Coulter), FoxP3+ (PE-Cy5, eBiosciences), IL-10 (FITC, R&D Systems, Minneapolis, MN, USA); cytotoxic T lymphocytes were identified as CD45+ (KO525, Beckman Coulter), CD8+ (PC7, Beckman Coulter), and CD107A+ (PE, Beckman Coulter).

**Monocyte subsets and MHC class II expression:** classical monocytes were identified as CD45+ (KO525, Beckman Coulter), CD14+ (PE-Cy7, Beckman Coulter), HLA-DRII (ECD, Beckman Coulter); non-classical monocytes as CD45+ (KO525, Beckman Coulter), CD16+ (PE-Cy5, Beckman Coulter), and HLA-DRII (ECD, Beckman Coulter); intermediate monocytes as CD45+ (KO525, Beckman Coulter), CD14+ (PE-Cy7, Beckman Coulter), CD16+ (PE-Cy5, Beckman Coulter), and HLA-DRII (ECD, Beckman Coulter).

PBMCs were incubated for 15 min with mAbs for cell surface antigen detection. Then, cells were permeabilized for 30 min with fixation/permeabilization buffer (eBiosciences) and further stained with antibodies for the detection of intracellular transcription factors and cytokines. Samples were acquired using a CytoFlex cytometer and data were analyzed using Kaluza software version 2.1.1 (Beckman Coulter).

### 2.5. Multiplex Cytokine Analyses

A 17-cytokine multiplex assay was performed on plasma, saliva, and cell culture supernatants from patients 10 h after PBMC stimulation as described above using a multiplexed magnetic bead immunoassay (Bio-Rad, Hercules, CA, USA) according to the manufacturer’s instructions via Luminex 100 technology (Luminex, Austin, TX, USA). Some of the resulting targets were over the range and an arbitrary value of 4000 pg/mL was assigned, while 0 pg/mL was attributed to values below the limit of detection.

## 3. Results

### 3.1. B Cell Subpopulations

The percentage of B lymphocyte subpopulations was comparable in CTR and PRS at baseline as well as after specific stimulations. Moreover, flu and anit-CD3/CD28 stimulations did not alter the percentage of plasmablast, transitional, naïve, and switched plasmablast B lymphocytes in either individual (Figure 1A–D). The only difference observed was in the unswitched-memory B cell subpopulation (Figure 1E). Indeed, both at baseline and upon Flu stimulation, a higher percentage of unswitched memory B cells was observed in the PRS patient compared to CTR. Conversely, anti-CD3/CD28 stimulation resulted in a 10% reduction in unswitched-memory B cells compared to the mock condition in the PRS patient alone, while we observed the opposite trend in the CTR subject.

### 3.2. T Cell Subpopulations

T cell analysis showed that the percentage of CD8+ T cells was higher in PRS compared to the CTR individual in both the treated and untreated condition (Figure 2A). Conversely, we observed an increased percentage of degranulating cells (CD8+CD107A+) of CD8+ T cells in CTR compared to the PRS patient mainly following flu stimulation (Figure 2B). Activated CD4+ (CD4+ HLADRII+) (Figure 2C), Treg (FOXp3-IL-10) (Figure 2D), and Th17 (CD4+Th17) (Figure 2E) T lymphocytes were significantly increased in the PRS patient compared to CTR following both flu and anti-CD3/CD28 stimulation.

### 3.3. Monocyte Subpopulations

The percentage of total monocytes was comparable in both patients (Figure 3A). In the unstimulated condition, a higher percentage of intermediate and non-classical monocytes was present in the PRS patient, whereas the percentage of classical monocytes was similar in the two individuals (Figure 3B–D). After flu and anti-CD3/CD28 stimulation, the percentage of non-classical monocytes was reduced in both PRS and CTR patients. The same reduction was also observed for intermediate monocytes in PRS with both stimulations, while we obtained the opposite result in CTR.

### 3.4. Cytokine/Chemokine Concentration in Cell Culture Supernatant, Plasma, and Saliva

We analysed cytokine and chemokine concentrations in PBMCs in the presence/absence of flu or anti-CD3/CD28 stimulation. Results showed that in all conditions, IL-1β, IL-17, and TNF-α concentrations were higher in the PBMC supernatant from PRS compared to the CTR patient. Conversely, the concentration of IL-6, IL-10, IFNγ, MCP1, MIP1α, and growth factors such as GMCSF (except for the anti-CD3/CD28 condition) and G-CSF was increased in CTR compared to PRS in all conditions. In both patients, stimulations increased cytokine release, mainly following anti-CD3/CD28 stimulation (Figure 4A,B).

The same cytokine/chemokine panel was used to analyze plasma and saliva samples. Only those cytokines/chemokines which were relevantly modulated are reported in Figure 5A–C. In particular, higher concentrations of plasmatic (ILRa, IL-7, IL-8, IL-9 TNFα) (Figure 5A) and salivary (IL-1β, ILRa and TNFα) (Figure 5B) pro-inflammatory cytokines and chemokines (plasma: CXCL10, MCP-1, MIP-1β, RANTES, Eotaxin) (Figure 5A) (saliva: CXCL-10, MCP-1, MIP-1α) (Figure 5B) were detected in CTR compared to the PRS patient, indicating the presence of an underlying immune activation in the CTR patient. Further confirming this assumption, a lower production of growth factors such as FGF and VEGF was detected both in the plasma (Figure 5C) and saliva (Figure 5D) of the PRS patient compared to the CTR patient.

## 4. Discussion

In this case–control study, we describe the immunological profile of two immunosuppressive therapy-naïve patients with a diagnosis of either Parry–Romberg syndrome (PRS) or scleroderma (CTR) and identify peculiar immunological differences between the two conditions.

Several reports have convincingly documented that dysregulated B cell function represents a hallmark of scleroderma diseases. Indeed, B cells have been found in lesional sites such as the alveolar interstitium and small blood vessels, and B lymphocyte subpopulations have been shown to be altered and to display an activated phenotype in scleroderma patients [11,12,13]. In our case–control study, B cell subpopulations were very similar in the PRS and CTR patients, except for the unswitched-memory B subset, which was more abundant in the PRS patient. Unswitched memory are antigen-experienced B cells expressing IgD surface protein [14]. Their contribution to humoral immunity remains controversial and additional work is needed to better understand their possible differentiation to autoreactive plasma cells in autoimmune diseases. Overall, a negative correlation between autoantibody concentrations and frequency of unswitched-memory B cells suggests that this subset may be protective against autoimmunity. Thus, lower percentages of unswitched-memory B cells are seen in systemic lupus erythematosus (SLE) and Sjögren’s syndrome (SS) patients, in whom higher concentrations of autoreactive antibodies are observed [15,16,17]. The increase in this B cell subpopulation in PRS might, therefore, represents an appropriate response to therapy, but further analyses should be performed in larger cohorts to draw the right conclusions.

Pathological conditions of systemic sclerosis include microvascular damage, inflammation, and immune abnormalities. Different T cell subtypes may cause vasculitis and fibrosis in scleroderma patients by up- and down-regulating cell surface molecules, altering the production of pro-fibrotic or pro-inflammatory cytokines, or by direct contact with fibroblasts [18,19,20]. The main T cell subtypes driving the pathogenesis of the disease include regulatory T Cells (Treg), interleukin-17 (IL-17)-producing Th17 cells, and CD8^+^ cytotoxic T lymphocytes (CTLs). Notably, the percentage of both CD8^+^ and CD4^+^ T cell subpopulations was different in PRS compared to the CTR patient. In particular, while the percentage of total CD8^+^ T cells was considerably higher, the percentage of degranulating cytotoxic CD8^+^ T cells, which are responsible for cytotoxicity, was reduced in PRS. It has been reported that the cytotoxic killing mediated by perforin and granzyme B can generate autoantigens that foster and/or prolong the immune response, as self-protein fragments generated by granzyme B are autoantibody targets in scleroderma disease [21]. Once more, further studies will be necessary to validate the results obtained and clarify the function of CD8^+^ T cells in PRS.

The percentage of all analyzed CD4^+^ T cell subsets was different in PRS compared to CTR. Indeed, the percentage of activated Treg and Th17 T lymphocytes was substantially greater in PRS compared to CTR, allowing for several speculations. In particular, the increased percentage of Th17, known to be involved in the pathogenesis of multiple autoimmune diseases, seems to be counterbalanced by the rise in Treg, which, contrariwise, protects from auto-aggression and tissue damage. Notably, our finding supports the use of anti-IL17 receptor monoclonal antibody in PRS patients, as suggested by Sideris et al., who successfully tested secukinumab in a PRS patient [22]. Another anti-IL17 receptor monoclonal antibody, brodalumab, is under investigation in two different clinical trials on systemic sclerosis (phase III clinical trial Clinicaltrials.gov identifier: NCT03957681; phase I clinical trial Clinicaltrials.gov identifier: NCT04368403). Notably, we have to point out that the low percentage of Th17 cells observed in our CTR patient is in contrast with data published by other researchers on scleroderma disease [23,24,25,26].

Monocyte subsets were comparable in PRS and CTR. Overall, in both PRS and CTR, a prompt response to stimuli was observed in classical monocytes, known to elicit a robust immune response and to be highly phagocytic and important scavenger cells. In contrast, our data showed that, following stimulation, intermediate and non-classical monocyte populations were reduced, mainly in the PRS patient.

The results of cytokine and chemokine production showed that both in the unstimulated and stimulated conditions, the secretome of PRS is characterized by a higher production of IL-1β, IL-6, and, mostly, IL-17, suggesting that a more robust immunoactivation is present in PRS. Moreover, analysis of the secretome of stimulated PBMCs showed that, although these patients are in an immunoactivated condition, their immune system is still able to react to external stimuli.

High concentrations of different, mostly pro-inflammatory, cytokines and chemokines were observed in the plasma and saliva of both individuals, suggesting that different degrees of immune activation are present both in PRS and scleroderma. Jacquerie et al. [27] reported an evident increase in several critical growth factors, including matrix metalloproteinases and chemokines (IGFBP-1, TGF-β, IL-8, YKL-40, and MMP-7) in induced sputum of scleroderma patients with pulmonary involvement, thus proposing sputum as a suitable and minimally invasive fluid to predict and monitor the evolution of the disease and treatment response. The suitability of sputum as a source of diagnostic and prognostic biomarkers has been investigated in other pathologies with dysregulated immunity. Nilsson et al. [28] demonstrated an increase in B-cell activating factor, IL-6, and IL-8 in induced sputum in a patient with SS, suggesting a specific ongoing inflammatory disease process in the airways of these patients. Notably, IL-8 and IL-6 were increased even in the saliva of the PRS patient enrolled in our study, further reinforcing the possibility of monitoring disease progression in saliva, although further studies are necessary to validate these results in larger cohorts.

Several clinical studies describe the clinical and neurological similarities of PRS and scleroderma patients. However, no one has compared their immunological profiles. Our case–control study is, therefore, the first to address this issue. According to our results, these two conditions are immunologically similar in basal condition, but differences emerge following specific stimulation. Further studies with larger numbers of patients are required to confirm our findings and verify if such differences may be exploited as diagnostic markers to better design future therapeutic approaches.

## 5. Study Limitations

The most significant limitation of this study is that, it being a case–control study, only two patients were enrolled, and they may not adequately represent the variability and diversity of PRS and scleroderma patients. This limitation affects the possibility to draw robust conclusions about the relationship between variables or the effectiveness of interventions.

On the other hand, given the low frequency of PRS and the lack of immunological studies profiling the disease, these findings can provide valuable insights into the molecular pathways governing the immunopathogenesis of the disease, which, in turn, could be exploited to properly set up future studies on larger cohorts.

Another limit of this study concerns a possible bias in the assessment of inflammatory cytokines, as both patients are under therapeutic approaches (CTR: iloprost infusion; PRS: valproic acid and lamotrigine) which could reduce cytokine production [29,30]. However, in both patients, PBMC stimulation resulted in an increased production of cytokines, suggesting that despite therapy, cytokine release is not compromised.

## Figures and Tables

**Figure 1 jcm-13-01219-f001:**
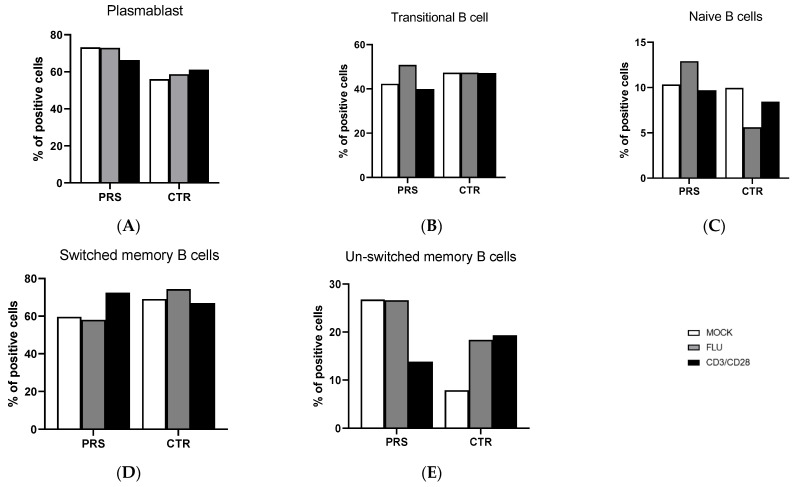
B cell subpopulation evaluation by flow cytometry. The percentage of plasmablast (**A**), transitional cells (**B**), naïve B cells (**C**), switched-memory B cells (**D**) and unswitched-memory B cells (**E**) was evaluated in PBMCs of PRS and CTR patients 10 h post flu (grey bar) and anti- CD3/CD28(black bar) stimulations and in mock condition (white bar).

**Figure 2 jcm-13-01219-f002:**
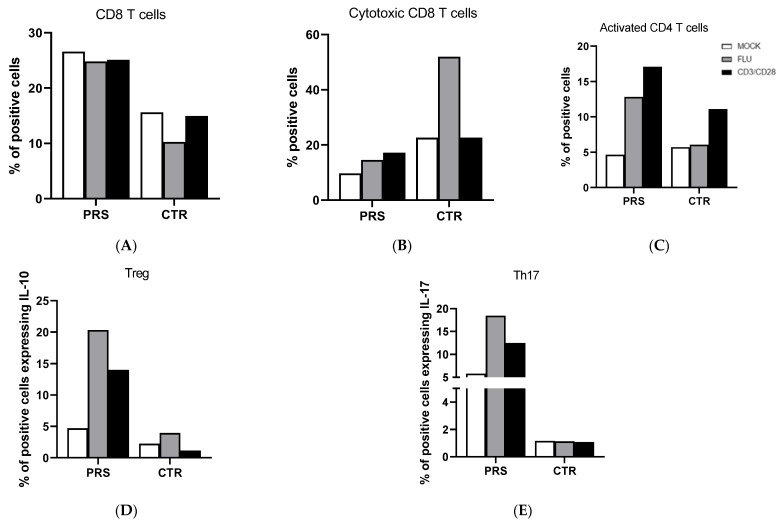
CD4+ and CD8+ T cell subset evaluation by flow cytometry. The percentage of CD8+ T cells (**A**), cytotoxic T cells (**B**), activated CD4+ T cells (**C**), Treg (**D**), and Th17 cells (**E**) was evaluated in PBMCs of PRS and CTR patients 10 h post flu (grey bar) and anti CD3/CD28 (black bar) stimulations and in mock condition (white bar).

**Figure 3 jcm-13-01219-f003:**
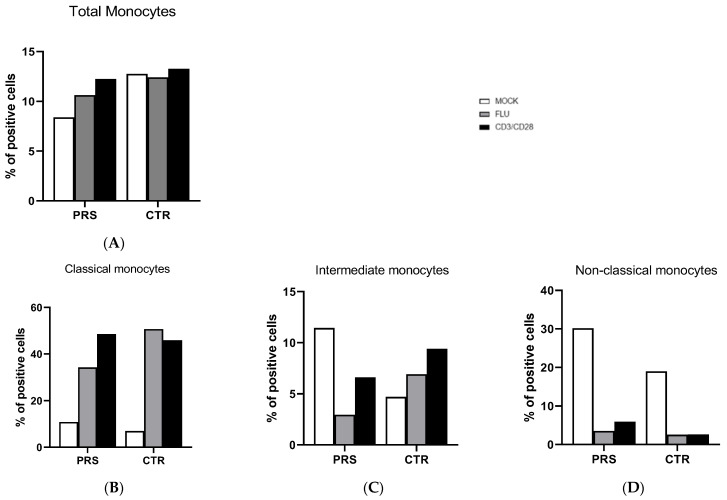
Monocyte subset evaluation by flow cytometry. The percentage of total monocytes (**A**), classical (**B**), intermediate (**C**), and non-classical (**D**) monocytes was evaluated in PBMCs of PRS and CTR patients 10 h post flu (grey bar) and anti CD3/CD28 (black bar) stimulations and in mock condition (white bar).

**Figure 4 jcm-13-01219-f004:**
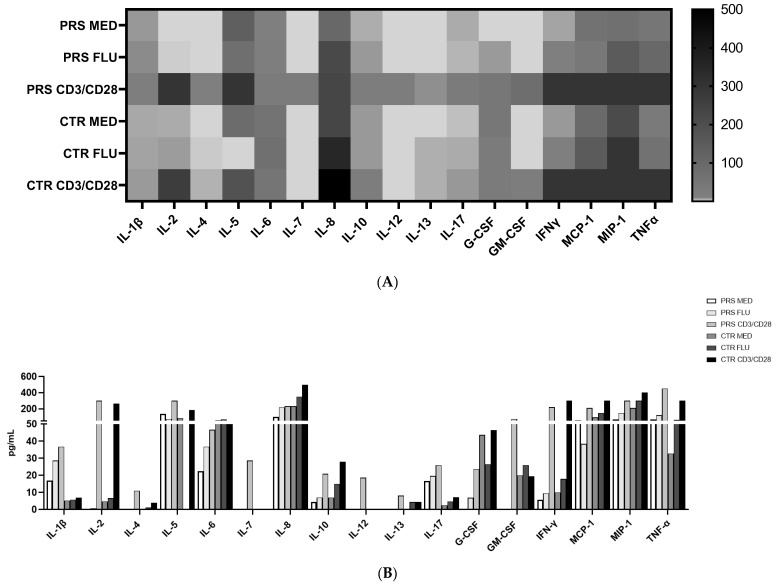
Cytokine/chemokine assessment by Luminex assay. (**A**) Heat map and (**B**) histogram representation of cytokine/chemokine production assessed by multiplex ELISA in PRS and CTR PBMC supernatants stimulated with flu and anti CD3/CD28.

**Figure 5 jcm-13-01219-f005:**
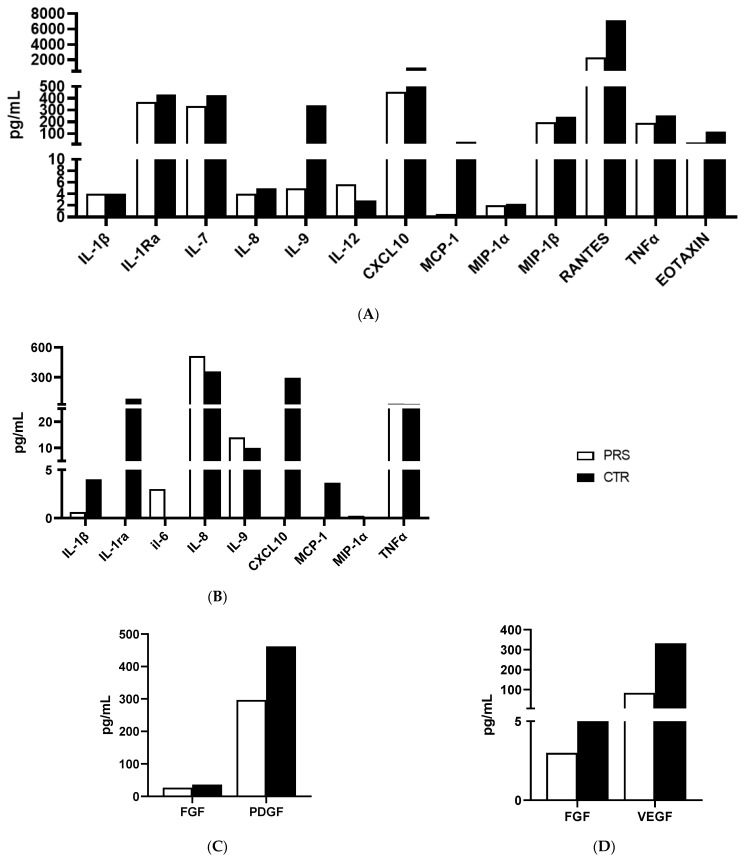
Cytokine/chemokine production assessment by Luminex assay. Analyses of cytokines/chemokines (**A**,**B**) and growth factors (**C**,**D**) was performed in plasma (**A**,**C**) and saliva (**B**,**D**) of PRS (white bar) and CTR (black bar).

**Table 1 jcm-13-01219-t001:** Clinical features of scleroderma (CTR) and Parry–Romberg syndrome (PRS).

Systemic Sclerosis	Early (<3 Years after Onset)	Late (>3 Years after Onset)
Prevalence	Between 8.5 and 85 in 250,000	
Age of onset (years)	47.3	
Female/male	Ranging from 3:1 to 8:1	
Related autoantibodies	Anti-nuclear antibody (ANA); anti-topoisomerase I (anti-Scl-70) antibody; anti-centromere antibody (ACA); anti-RNA polymerase III antibody; anti-Th/To antibody; U1 RNP antibody; U3 RNP (fibrillarin) antibody; PM-Scl antibody; anti-U11/U12 antibody	
Constitutional	Fatigue and weight loss	Minimal, weight gain typical
Vascular	Raynaud often relatively mild	Raynaud more severe, more telangiectasia
Cutaneous	Rapid progression involving arms, trunk, face	Stable or regression
Musculoskeletal	Prominent arthralgia, stiffness, myalgia, muscle weakness, tendon friction rubs	Flexion contractures and deformities, joint/muscle symptoms less prominent
Gastrointestinal	Dysphagia, heartburn	More pronounced symptoms, midgut and anorectal complications more common
Cardiopulmonary	Maximum risk of myocarditis, pericardial effusion, interstitial pulmonary fibrosis	Reduced risk of new involvement but progression of existing established visceral fibrosis
Renal	Maximum risk of renal crisis within the first 5 years	Renal crisis less frequent, uncommon after 5 years
Treatment	Methotrexate (MTX), mycophenolate mofetil (MMF), intravenous immune globulin (IVIG), rituximab, tocilizumab, cyclophosphamide	
**Parry–Romberg syndrome**	**Early (<10 years after onset)**	**Late (>10 years after onset)**
Prevalence	1 in 250,000	
Age of onset (years)	13.6	
Female/male	3:01	
Related autoantibodies	Anti-nuclear antibody (ANA); anti-RNP antibody; rheumatoid factor (RF); anti-histone antibody; anti-centromere antibody	
Constitutional	None	Weight loss only secondary to neurological and oral complications
Vascular	Raynaud rare	Raynaud rare
Cutaneous	Normal to hyperpigmented skin, normal-appearing hair with minimal or no skin induration. Dermal sclerosis with inflammation in early stages	Progressive hemifacial atrophy of soft and hard tissue of one side of face, usually left, frequently associated with ocular pathology. Late stage with fat atrophy and relative shrinking of adnexa at later stages
Musculoskeletal	Mostly lower half of the face, predominantly maxilla and mandibular regions	Otorhinolaryngological, oral, and dental
Gastrointestinal	Usually no involvement	Dysphagia secondary to neurological disease
Cardiopulmonary	Usually no involvement	Usually no involvement
Renal	No involvement	Usually no involvement
Treatment	Topical calcineurin inhibitors, methotrexate (MTX), mycophenolate mofetil (MMF), azathioprine, corticosteroids, hydroxychloroquine	

## Data Availability

Data are available upon request to corresponding author.

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
