# Peer review of "Immunological Profiles in Parry–Romberg Syndrome: A Case–Control Study"

_jcm, 2024, doi:10.3390/jcm13051219_

Round 1

Reviewer 1 Report

Comments and Suggestions for Authors

Dear Authiors and Editors!

Thank you for the opportunity to review the manuscript.

PRS is an ultra-rare disease with overlap features with systemic scleroderma. The etiology is unknown and a lot of manuscript does not support the anti-rheumatic treatment. This manuscript has an absolutely new poitnof this disease with promising treatment. RThe Authors studied immunonolical profile of patient with PRS and compared with classical localized scleroderma. 

Tjhe manuscript is highly actual. The methods are modern. The results are clear and discussion contains necessary contemporary literature.

I can suggest Authors to organize the limitation subsection because it is a single case on=bservation.

Also will be very nice if Authors add the table in the discussion with short description of both disease (comparison) and information about possible treatment with biologics of PRS or both. It will be useful for readers. 

Author Response

Reviewer 1: 

 I can suggest Authors to organize the limitation subsection because it is a single case on=bservation.

We thank the reviewer for her/his suggestion that can significantly improve the quality of the manuscript. Therefore, at the end of the manuscript we added a paragraph reasoning on the limitations of the study: “The most relevant limitation of the study is that being a case-control study only two patients were enrolled and they may not adequately represent the variability and diversity of PRS and Scleroderma patients. This limitation affects the possibility to draw robust conclusions about the relationship between variables or the effectiveness of interventions.

 On the other hand, given the low frequency of PRS and the lack of immunological studies profiling the disease these findings can provide valuable insights into the molecular pathways governing the immunopathogenesis of the disease which in turn, could be exploited to properly set-up future studies on larger cohorts.” Pag. 12 line 325-332

Also will be very nice if Authors add the table in the discussion with short description of both disease (comparison) and information about possible treatment with biologics of PRS or both. It will be useful for readers. 

We thank Reviewer 1 for her/his insightful suggestion. Accordingly, we add a table describing the epidemiologic and clinical features of CTR and PRS (Page 2, Table 1).

As for the request to discuss possible biological treatments, we prefer to exclusively discuss only those treatments that have been previously tested and documented in the literature, such as IL-17 (phase III clinical trial Clinicaltrials.gov identifier: NCT03957681; phase I clinical trial Clinicaltrials.gov identifier: NCT04368403). Indeed, we believe that dissecting the role of biological treatments which effects have not been scientifically proved and published could be confusing for the reader.

Reviewer 2 Report

Comments and Suggestions for Authors

Comments:

 1.      Abstract: Standard abbreviation to be used for. e.g. scleroderma (CTR), Flu- ? (Flu- or antiCD3/CD28-stimulated PBMCs) (expand it when first used to avoid confusion ), scleroderma patients or patient?  

2.      Introduction, what is the rationale of comparing immunological parameters of PRS with patient affected by scleroderma (CTR)? It is not justified in introduction section. Why not healthy control taken for comparison.

3.      Line#51 ‘(eg, lupus),’ line#77, line #93, ‘One × 106 PBMCsprostaglandin I2 use standard abbreviation/appropriate use of the sub-super-script; line#270, IL-1b. Manuscript must be proof read appropriately to avoid such mistakes.

4.      Mention the gating strategy adopted to characterize different immune cell population by flow cytometry. What control used for gating. How doublets excluded. Methodology poorly described.

5.      Authors stimulated PBMCs with two different stimuli, 1) combination of two viruses (two live 93 UV-inactivated influenza viruses (Flu): an influenza A virus (A/RX73 and A/Puerto 94 Rico/8/34 strains; 1:800) and the 1998–1999 formula of flu vaccine (1:5000)) and 2) 1ug/mL of anti-CD3/CD28. What is the rationale of using two stimuli?

6.      Differences in cell percentage will be not relevant unless compared with healthy control. Fortifying the datasets with healthy control data is highly recommended. Although relevance of such comparison is questionable as it is statistically not acceptable. Further, comparing the two disease conditions without knowing the normal control values, is of less importance.

7.      Section 3.4, ‘Results showed that in all conditions IL-1β, IL-6, IL-17, and TNF-α concentration was higher in supernatants of PBMC from the PRS’ data not supporting the statement, give bar graph presentation of the as 3.4(b) and keep heatmap presentation as 3.4(a).

8.      Line#199, ‘Higher concentration of salivary (IL-1β, ILra, TNFα) (Figure 3.5A)’ in figure 3.5A, IL-1β is comparable in both group with no difference. There are lots of discrepancies in data and text. Authors should proof read the entire manuscript for such errors. All cytokines mentioned in supernatant of PBMCs not studied in saliva and plasma? Why? Justification required.

9.      PRS patient taking medication i.e. valproic acid, clobazam and lamotrigine and CTR patient taking prostaglandin I2 analog (iloprost). There are studies where prostaglandin I2 analog can modulate cytokine levels for e.g. Kuo et al., Mol Med . 2012 May 9;18(1):433-44. doi: 10.2119/molmed.2011.00193. Similarly, modulatory effect of valproic acid, clobazam and lamotrigine can be debated. Authors, should cite such studies and discuss the findings with their result.

10.  Typos, punctuation, fonts require special attention.

Comments on the Quality of English Language

English require attention. Refer to above comments.

Author Response

Reviewer 2: 

  1. Abstract: Standard abbreviation to be used for. e.g. scleroderma (CTR), Flu- ? (Flu- or antiCD3/CD28-stimulated PBMCs) (expand it when first used to avoid confusion ), scleroderma patients or patient?  

We thank the reviewer for her/his attentive reading of our work and for the constructive criticisms. We do apologize for the carelessness in the abstract. Accordingly, we have expanded all the acronyms and amended the typos in the abstract (Pag. lines 16-19 and 21-22).

  1. 2.      Introduction, what is the rationale of comparing immunological parameters of PRS with patient affected by scleroderma (CTR)? It is not justified in introduction section. Why not healthy control taken for comparison.

The reviewer is right as reasons for comparing PRS and scleroderma patient should be more exhaustively cleared. That’s why, even following suggestion by Reviewer 1, we decided to add a table describing similarities and differences of the two pathologies (Table 1 Pag 2-4).  The reasons leading us to compare the immunologic profile of the two patients relies on the observation that Parry-Romberg syndrome and scleroderma are often matched due to their autoimmune nature and involvement of connective tissues. To stress the rationale supporting this comparison the following sentences were added in the introduction section:” Despite affecting different parts of the body and presenting with distinct symptoms, these two diseases share underlying pathological mechanisms related to autoimmunity and tissue fibrosis. Based on these premises”. (Pag 2 line 59-62).

“Comparing these two conditions could help clinicians to understand the spectrum of autoimmune diseases and to develop better strategies for diagnosis, management, and treatment. Additionally, studying immunological similarities and differences between PRS and CTR may provide insights into the pathogenesis of autoimmune disorders, leading to the identification of specific bio-markers possibly exploitable to promptly diagnose PRS and/or as new therapeutic targets”.  (Pag 2 line 65-71)

  1. Line#51 ‘(eg, lupus),’ line#77, line #93, ‘One × 106 PBMCs’prostaglandin I2 use standard abbreviation/appropriate use of the sub-super-script; line#270, IL-1b. Manuscript must be proof read appropriately to avoid such mistakes.

We do apologize for the mistakes and agree with the reviewer. The manuscript was carefully proofread and changes were made throughout the text and highlighted

  1. Mention the gating strategy adopted to characterize different immune cell population by flow cytometry. What control used for gating. How doublets excluded. Methodology poorly described.

We are thankful to the Reviewer for the suggestion. We have added the Supplementary Figure 1 in which the gating strategy for the principal subpopulations analysed and the criteria for doublets exclusion are summarized.  In the materials and methods section flow cytometry analysis we add this sentence to introduce the principal gaiting strategies to improve the methodology. “The gaiting strategies of the main lymphocytes subpopulation are shown in Supplementary Figure 1.”Pag. 5 line 115-116

  1. Authors stimulated PBMCs with two different stimuli, 1) combination of two viruses (two live 93 UV-inactivated influenza viruses (Flu): an influenza A virus (A/RX73 and A/Puerto 94 Rico/8/34 strains; 1:800) and the 1998–1999 formula of flu vaccine (1:5000)) and 2) 1ug/mL of anti-CD3/CD28. What is the rationale of using two stimuli?

We thank the reviewer for this question. In the section on cell culture conditions, we insert the following sentence to better explain the choice of stimulators. “To evaluate the global responsiveness and functionality of the immune system of both patients, 1× 106 PBMCs were stimulated with recall antigens (flu vaccine) and 1mg/mL of antiCD3 (Invitrogen Waltham, Massachusetts, United States)/CD28 (Biosigma, Venice, Italy). In particular, Flu stimulus results from the mixing of two UV-inactivated influenza viruses: an influenza A virus (A/RX73 and A/Puerto Rico/8/34 strains; 1:800) and the 1998–1999 formula of flu vaccine (1:5000; Wyeth Laboratories Inc., Marietta, PA, USA)”. (Pag. 5 line 105-111)

  1. Differences in cell percentage will be not relevant unless compared with healthy control. Fortifying the datasets with healthy control data is highly recommended. Although relevance of such comparison is questionable as it is statistically not acceptable. Further, comparing the two disease conditions without knowing the normal control values, is of less importance.

We do appreciate the thoughtful comment of the reviewer on the possibility to assess the same parameters on a healthy subject to be compared with the two patients. I would like to provide further context regarding the design and objectives of my study. The primary aim of our research is to discriminate between two distinct pathological conditions, specifically Perry Romberg Syndrome (PRS) and scleroderma, from an immunological standpoint. Given the rarity of PRS and the well-documented immunological features of scleroderma in the existing literature ( doi: 10.1186/s13075-022-02889-5; doi: 10.1097/BOR.0000000000000961; doi: 10.3389/fimmu.2022.999008.), the study focuses on elucidating the unique immunological profile associated with each condition. In this context, the inclusion of healthy controls may not align with the primary goal of the research, which is to differentiate between these two specific pathological entities. The comparative analysis between PRS and scleroderma aims to highlight the immunological distinctions between them rather than establishing a baseline with healthy controls. However, we do understand your concern about the statistical robustness of such comparison which affects the possibility to draw robust conclusions about the relationship between variables or the effectiveness of interventions. Such considerations have been added in a Study limitation paragraph at the end of the manuscript. Pag.12 line 325-337

  1. Section 3.4, ‘Results showed that in all conditions IL-1β, IL-6, IL-17, and TNF-α concentration was higher in supernatants of PBMC from the PRS’ data not supporting the statement, give bar graph presentation of the as 3.4(b) and keep heatmap presentation as 3.4(a).

As suggested by the reviewer in figure 3, we have kept the heatmap as 3.4(a) and added the corresponding histogram as 3.4(b). We do apologize for the error with IL-6, which production is definitely more elevated in the CTR. For this reason, the previous sentence was modified accordingly: “Results showed that in all conditions IL-1β, IL-17, and TNF-α concentration was higher in PBMC supernatant from the PRS compared to the CTR patient. Conversely, the concentration of IL-6, IL-10, IFNγ, MCP1, MIP1a and growth factors as GMCSF, (except for anti CD3/CD28 condition) and G-CSF was increased in CTR compared to PRS in all conditions. In both patients, stimulations increased the cytokine release, mainly following anti-CD3/CD28 stimulation”. (Pag. 8 line 203-208).  The figure legend was modified as well: Cytokines/chemokine production assessment by Luminex assay.  (A) Heat Map and (B) Histogram representation of cytokine/chemokine production assessed by multiplex ELISA on PRS and CTR PBMC supernatant stimulated with Flu and anti CD3/CD28. “(Pag. 9 line 218-220).

  1. 8.       Line#199, ‘Higher concentration of salivary (IL-1β, ILra, TNFα) (Figure 3.5A)’ in figure 3.5A, IL-1β is comparable in both group with no difference. There are lots of discrepancies in data and text. Authors should proof read the entire manuscript for such errors.

We do apologize to the reviewer, but there was a mistake in the figure legend. Panel 3.5A and C show the cytokines and growth factors quantified in the plasma of both patients. Indeed, data description in the manuscript is correct, while figure legend was wrong. For this reason, we have changed the figure legend (Pag. 9 line 238) .

  1. All cytokines mentioned in supernatant of PBMCs not studied in saliva and plasma? Why? Justification required

The same multiplex assay described in the Materials and Methods section was used to assess cytokine/chemokine production in plasma, saliva and PBMC supernatant. However, in the results describing cytokine/chemokine production in saliva and plasma, only those factors that were modulated in at least one of the two anatomical districts where reported in the histograms. To improve the comprehension of the text the following sentence was added into the manuscript:” The same cytokine/chemokine panel was used to analyze plasma and saliva samples. Only, those cytokines/chemokines which were relevantly modulated were reported in Figure 3.5 A-C.” (Pag 9 line 221-223)

  1. 9.      PRS patient taking medication i.e. valproic acid, clobazam and lamotrigine and CTR patient taking prostaglandin I2 analog (iloprost). There are studies where prostaglandin I2 analog can modulate cytokine levels for e.g. Kuo et al., Mol Med . 2012 May 9;18(1):433-44. doi: 10.2119/molmed.2011.00193. Similarly, modulatory effect of valproic acid, clobazam and lamotrigine can be debated. Authors, should cite such studies and discuss the findings with their result.

The reviewer's observation is pertinent. We add a paragraph about this limitation and we reported the suggested papers. Additionally, we came across studies supporting a potential decrease in inflammatory cytokines with both valproic acid and lamotrigine treatment, while there is no evidence in the literature for an interaction between clobazam and cytokines. A sentence stressing this limit has been introduced in the study limitation section: “Another limit of the study concerns a possible bias in the assessment of inflammatory cytokines as both patients are under therapeutic approaches (CTR: Iloprost infusion; PRS: Valproic Acid and Lamotrigine) which could reduce cytokine production [27, 28]. However, in both patients PBMC stimulation resulted in an increased production of cytokines, suggesting that despite therapy cytokine release is not compromised.”  Pag. 12 line 333-337

Inizio modulo

  1. Typos, punctuation, fonts require special attention.

According to reviewer 2 suggestion the manuscript was revised for font, typos and punctuation mistakes. All the identified errors have been traced all throughout the paper.